# Design and Implementation of a Hydroponic Strawberry Monitoring and Harvesting Timing Information Supporting System Based on Nano AI-Cloud and IoT-Edge

**Sun Park \*** and **JongWon Kim**

Artificial Intelligence Graduate School, Gwangju Institute of Science and Technology, 123 Cheomdangwagi-ro, Buk-gu, Gwangju 61005, Korea; jongwon@gist.ac.kr

\* Correspondence: sunpark@gist.ac.kr; Tel.: +82-62-715-6380

**Abstract:** The strawberry market in South Korea is actually the largest market among horticultural crops. Strawberry cultivation in South Korea changed from field cultivation to facility cultivation in order to increase production. However, the decrease in production manpower due to aging is increasing the demand for the automation of strawberry cultivation. Predicting the harvest of strawberries is an important research topic, as strawberry production requires the most manpower for harvest. In addition, the growing environment has a great influence on strawberry production as hydroponic cultivation of strawberries is increasing. In this paper, we design and implement an integrated system that monitors strawberry hydroponic environmental data and determines when to harvest with the concept of IoT-Edge-AI-Cloud. The proposed monitoring system collects, stores and visualizes strawberry growing environment data. The proposed harvest decision system classifies the strawberry maturity level in images using a deep learning algorithm. The monitoring and analysis results are visualized in an integrated interface, which provides a variety of basic data for strawberry cultivation. Even if the strawberry cultivation area increases, the proposed system can be easily expanded and flexibly based on a virtualized container with the concept of IoT-Edge-AI-Cloud. The monitoring system was verified by monitoring a hydroponic strawberry environment for 4 months. In addition, the harvest decision system was verified using strawberry pictures acquired from Smart Berry Farm.

**Keywords:** strawberry; cultivation environment monitoring; harvest decision; IoT edge; AI cloud; deep learning; hydroponic

## 1. Introduction

Strawberry cultivation is one of the most globalized horticultural industries, as strawberries are extensively consumed worldwide. The global strawberry market size is expected to reach USD 22,450 million by 2026, from USD 18,370 million in 2020. In other words, it is expected to grow at an annual average of 3.4% from 2021 to 2026. According to the Food and Agriculture Organization (FAO), worldwide strawberry production increased by 39.4% between 2008 and 2018. Strawberry production is increasing every year with increasing consumption [1,2].

Harvest maturity is an important factor in determining the shelf life and the quality, taste, juice and texture of the final fruit. Harvesting of immature strawberries leads to poor quality, internal deterioration and easy spoilage. Conversely, delayed harvesting of strawberries significantly increases fruit damage, which can lead to rapid losses after harvest. To manage the quantitative or qualitative loss of strawberries before and after harvesting, it is important to understand the delicacy of strawberries, conditions of physiological maturity, timely harvesting methods and other factors. The degradation of the quality of strawberries is a major problem for strawberry growers. Monitoring the growth and environment of strawberries can reduce damage to strawberries during harvest or

help farmers to control ripening progress. In general, the detection of all types of diseases and evaluation of the ripening stage of strawberries are carried out through manual inspection and evaluation according to the personal experience of the farmer. Manual identification of mature strawberries for harvesting is time- and labor-intensive. At the same time, strawberry production has to overcome unfavorable agricultural conditions, such as water shortages, changes in the growing environment and climate change. Thus, improving various strawberry farming practices with innovative technologies can enhance the strategic advantage of agricultural production.

In order to overcome such obstacles to strawberry farming, this paper designs and develops a system capable of monitoring the strawberry cultivation environment in real time and supporting decision makers with enhanced information about harvesting timing. The proposed system is designed to collect, store and analyze strawberry environmental data and photos with the concept of IoT-Edge-AI-Cloud [3,4]. The proposed monitoring system collects 13 types of related cultivation environment data using the IoT-Edge module. The collected environmental data are stored and visualized in a nano-sized private cloud-based database server and visualization server, respectively. The proposed harvest decision system classifies strawberry objects according to their maturity level using a classification model based on a deep learning YOLO (You Only Look Once) algorithm in a nano-sized private AI-Cloud-based analysis server. The IoT-Edge device of the proposed system can be implemented cost-effectively based on Arduino and Raspberry Pi. The proposed system can also easily and flexibly expand the system via container virtualization of the system based on AI-Cloud, even if the applied strawberry farm area increases. In other words, the number of container servers based on virtualization can easily be increased whenever necessary. The system can efficiently manage strawberry cultivation and harvest by integrating and visualizing strawberry cultivation environment monitoring data and strawberry ripeness classification. The monitoring and analysis results are visualized in an integrated interface, which provides a variety of basic data (e.g., production volume, harvest time and pest diagnosis) for strawberry cultivation. The proposed monitoring system can easily and stably construct big data of strawberry cultivation environments. In other words, it can monitor the 13 types of collected environmental data in real time to understand the growth environment. Additionally, the optimal growth environment can be analyzed by utilizing big data that have been accumulated from the growth cycle. The remainder of this paper is organized as follows: Section 2 describes the related studies on methods of growing environment monitoring and maturity classification; Section 3 explains the hydroponic strawberry monitoring and harvest decision system; Section 4 describes the operational use cases of the hydroponic strawberry monitoring and the tests of the harvest decision system. Finally, Section 5 discusses the conclusions and future direction.

## 2. Related Works

This chapter reviews the related studies on fruit and vegetable cultivation monitoring and maturity classification. Bharti et al. [5] proposed a hydroponic tomato monitoring system that uses a microprocessor to transmit temperature and plant size data to the cloud using the message queuing telemetry transport (MQTT). It can also check the saved data via an Android application. Joshitha et al. [6] used Raspberry Pi and sensors to store data on temperature, humidity, water level, soil level, etc., in the Ubidots cloud database from a hydroponic cultivation system. Herman and Surantha [7] proposed an intelligent monitoring and control system for hydroponic precision farming. The system was used to monitor the water and nutrition needs of plants, while fuzzy logic was designed to precisely control the supply of water and nutrients. Fakhrurroja et al. [8] proposed an automatic pH (potential of hydrogen) and humidity control system for hydroponic cultivation using a pH sensor, humidity sensor, Arduino, Raspberry Pi, and fuzzy logic. As a result of the fuzzy model, the pH of the water is controlled using a nutrient pump and a weak acid pump. Verma et al. [9] proposed a framework to predict the absolute crop growth rate using a machine learning method for the tomato crop in a hydroponic

system. Their method helps to understand the impact of important variables in the correct nutrient supply. Pawar et al. [10] designed an IoT-enabled Automated Hydroponics system using NodeMCU and Blynk. Their method consists of a monitoring stage and automation. In the monitoring stage, temperature, humidity and pH are monitored. During automation, the levels of pH, water, temperature and humidity are adjusted. Issarny et al. [11] introduced the LATTICE framework for the optimization of IoT system configuration at the edge, provided the ontological description of the target IoT system. The framework showed an example applied to a hydroponic room of vegetables for monitoring and controlling several physical variables (e.g., temperature, humidity, $CO_2$, air flow, lighting and fertilizer concentration, balance and pH). Samijayani et al. [12] implemented wireless sensor networks with Zigbee and Wi-Fi for hydroponics plants. The networks are used by the Zigbee-based transceiver in the sensor node and the Wi-Fi-based gateway in the coordinator node. Adidrana and Surantha [13] proposed a monitoring system to measure pH, TDS (total dissolved solids) and nutrient temperature values in the nutrient film technique using a couple of sensors. The system used lettuce as the object of experiments and applied the k-nearest neighbor algorithm to predict the classification of nutrient conditions.

Ge et al. [14] presented a machine vision system in a strawberry-harvesting robot for the localization of strawberries and environment perception in tabletop strawberry production. The system utilized a deep learning network for instance segmentation to detect the target strawberries. An environment perception algorithm was proposed to identify a safe manipulation region and the strawberries within this region. A safe region classification method was proposed to identify the pickable strawberries. Yu et al. [15] proposed a harvesting robot for ridge-planted strawberries and a fruit pose estimator. The proposed harvesting robot was designed on the servo control system of a strawberry-harvesting robot suitable for the narrow ridge-planting mode. The fruit pose estimator, based on the rotated YOLO (R-YOLO), was suitable for strawberry fruit in the narrow spaces of the ridge-planting mode. Feng et al. [16] designed a harvesting robot for tabletop-cultivated strawberry. The robot system consists of an information acquisition part, a harvesting execution part, a controller and other auxiliaries. The information acquisition part includes a distant- and close-view camera, an artificial light source and obstacle detection sensors. The distant-view camera is used to dynamically identify and locate the mature fruit in the robot's view field. The close-range camera is used to obtain close-view images of the fruit. The artificial light source can compensate for the variable sunlight conditions under agricultural environments. Huang et al. [17] proposed a fuzzy Mask R-CNN (regions with convolutional neural network) model to automatically identify the ripeness levels of cherry tomatoes. The proposed method used a fuzzy c-means model to maintain the spatial information of various foreground and background elements of the image. It also used Mask R-CNN to precisely identify each tomato. The method used a hue saturation value color model and fuzzy inference rules to predict the ripeness of the tomatoes. Altaheri et al. [18] proposed a machine vision framework for date fruit harvesting, which uses three classification models to classify date fruit images in real time according to their type, maturity and harvesting decision. Zhang et al. [19] proposed a CNN-based classification method for a tomato-harvesting robot to improve the accuracy and scalability of tomato ripeness with a small amount of training data. The authors of [20] proposed a scheme using machine-vision-based techniques for automated grading of mangoes according to their maturity level in terms of actual days to rot and quality attributes such as size and shape. Saputro et al. [21] introduced a banana maturity prediction system using visible near-infrared imaging based on the chlorophyll characteristic to estimate maturity and chlorophyll content non-destructively. Kuang et al. [22] proposed a kiwifruit classifier using a multivariate alternating decision tree and deep learning.

The proposed system collects 13 types of strawberry growth environment data, whereas the environmental information monitored in the previous related works [5–13] consisted of two to eight types. By collecting more environmental information compared

to the related works, it is possible to access more diverse methods when analyzing the growing environment. As the related works [5–13] involve simply collecting environmental data or simply storing the collected data in the cloud, the addition of functions to the related works is limited. By contrast, the proposed method is designed to facilitate function expansion and analysis, as it consists of an IoT-Edge module and a nano-sized private AI-Cloud module. The proposed method determines the strawberry harvest time by using a deep-learning-based method similar to related studies [14,15,17,19]. However, the difference from the related studies is that it is designed based on a virtualized container to increase the scalability of the function.

## 3. Hydroponic Strawberry Monitoring and Harvest Decision System

The following sections describe the components of the device hardware and module architecture for the hydroponic strawberry monitoring and harvest decision system.

### 3.1. System Overview and Device Components

A hydroponic strawberry monitoring and harvest decision system prototype was designed for collecting growth environment data, analyzing optimal growing environments and identifying mature strawberries. The proposed system consists of a hydroponic strawberry monitoring IoT-Edge device and a GPU workstation device, as shown in Figure 1.

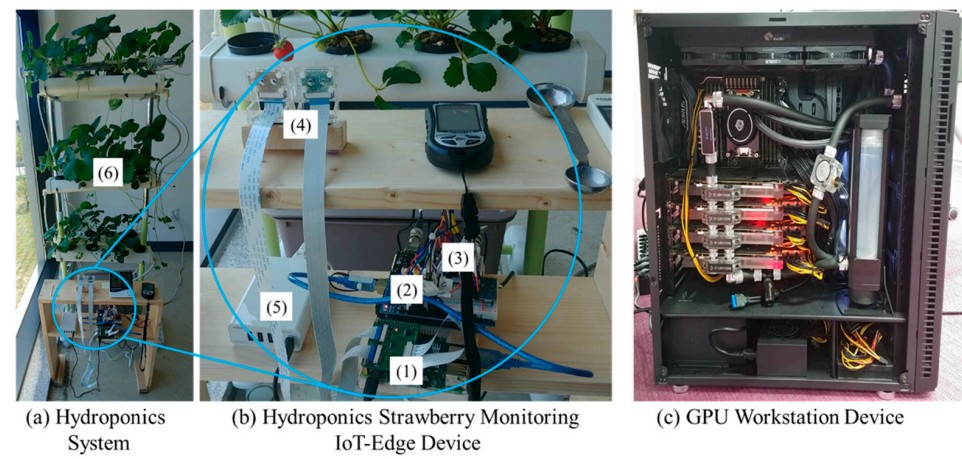

**Figure 1.** The hydroponic strawberry monitoring and harvest decision system.

### 3.1.1. Hydroponics System

In this paper, a home hydroponic cultivation system was used to monitor strawberry growth information and verify harvest decision information, as shown in Figure 1(6). The hydroponic cultivation system consists of a hydroponic shelf, a water tank and a water pump. The hydroponic shelf consists of a total of three floors with two pipes per floor and 32 plants. Seolhyang strawberry [23] was grown in the hydroponic cultivation system.

### 3.1.2. Hydroponic Strawberry Monitoring IoT-Edge Device

The IoT-Edge device collects growth environment data for optimal cultivation environment analysis when growing the hydroponic strawberry solution. In addition, strawberry images are taken for identification of mature strawberries and calculation of the normalized difference vegetation index (NDVI). The IoT-Edge device is composed of Raspberry Pi, Arduino, sensors, Raspberry Pi cameras and a power supply, as shown in Figure 1b. Table 1 shows the component hardware specification of the IoT-Edge device. In this study, environmental data and strawberry photos were collected in a fixed environment for verification of the proposed system. For this reason, one strawberry plant was selected and the distance between the camera and the strawberry plant was fixed. The Raspberry Pi Camera v2.1 module has a fixed focus, so we manually turned the focus ring to focus. The camera

module in Figure 1(4) is connected to the Arducam multi-camera module that is connected to the Raspberry Pi module in Figure 1(1).

**Table 1.** The component hardware specification of the IoT-Edge device.

| Component | Hardware Specification |
|---|---|
| Raspberry Pi (Figure 1(1)) | • Raspberry Pi 3B+<br>   - CPU: ARM Cortex-A53 1.4 GHz<br>   - RAM: 1 GB SRAM<br>   - Wi-Fi: 2.4 GHz and 5 GHz<br>   - Ethernet: 300Mbps<br>• microSD 256 GB<br>• Arducam Multi Camera Adapter Module V2.1<br>   - Work with 5 MP or 8 MP cameras<br>   - Accommodate 4 Raspberry Pi cameras on a single RPi board<br>   - 3 GPIOs required for multiplexing<br>   - Cameras work sequentially, not simultaneously |
| Arduino (Figure 1(2)) | • Arduino Mega 2560<br>   - Microcontroller: ATmega2560<br>   - Digital I/O pins: 54<br>   - Analog input pins: 16<br>   - Flash memory: 256 KB<br>   - SRAM: 8 KB<br>   - EEPROM: 4 KB<br>   - Clock speed: 16 MHz |
| Sensors (Figure 1(3)) | • Light intensity sensor (lux): GY-30<br>• pH sensor: SEN0161 (pH probe, circuit board, analog cable)<br>• Dissolved oxygen sensor: Kit-103DX (DO circuit, probe, carrier board)<br>• Ultraviolet sensor: ML8511<br>• TDS Sensor: Gravity TDS Meter v1.0 (EC (electrical conductivity), TDS (total dissolved solids))<br>• Temperature/Humidity/Pressure/Altitude: BME/BMP280<br>• Water temperature sensor: DS18B20<br>• $CO_2$ sensor (value, status): MG811 |
| Cameras (Figure 1(4)) | • Raspberry PI Camera Module V2.1/Raspberry PI NoIR Camera Module V2.1 (8 megapixel)<br>   - 3280 × 2464 resolution<br>   - CSI (camera serial interface)-2 bus<br>   - Fixed focus module |
| Power supply (Figure 1(5)) | • USB Smart Charger 5v 2A 5 ports |

### 3.1.3. GPU Workstation

The GPU workstation device in Figure 1c stores the growing-environment data and strawberry images collected from the IoT-Edge device. It also selects images of mature strawberries that can be harvested. The NDVI value is calculated to determine if the strawberry is healthy. Table 2 shows the component hardware specification of the GPU workstation device.

**Table 2.** The component hardware specification of the GPU workstation device (Figure 1c).

| Component | Hardware Specification |
|---|---|
| CPU | • AMD Ryzen Threadripper 2950X<br>  - 16-Core Processor 32 Thread<br>  - 3.5 GHz (4.4 GHz Max Boost)<br>• Water cooling system |
| RAM | • Samsung DDR4<br>  - 16 GB * 8 = 128 GB<br>  - Configuration clock speed: 2666 MT/s |
| SSD | • m.2 NVMe 1TB |
| Main board | • X399 AORUS PRO<br>  - Supports AMD 2nd Generation Ryzen™ Threadripper™<br>  - Quad Channel ECC/Non-ECC DDR4, 8 DIMMs<br>  - Fast Front and Rear USB 3.1 Type-C™ Interface<br>  - 4-Way Graphics Support |
| GPU | • ASUS ROG STRIX GTX 1080ti * 4<br>  - Base Clock 1596 MHZ<br>  - Core 3584<br>• Water cooling system |

*3.2. System Module Architecture*

The system module architecture was designed to collect, store and analyze strawberry cultivation environment information from a software point of view for functions' implementation. The system module consists of a monitoring IoT-Edge module and an analysis station module, as shown in Figure 2.

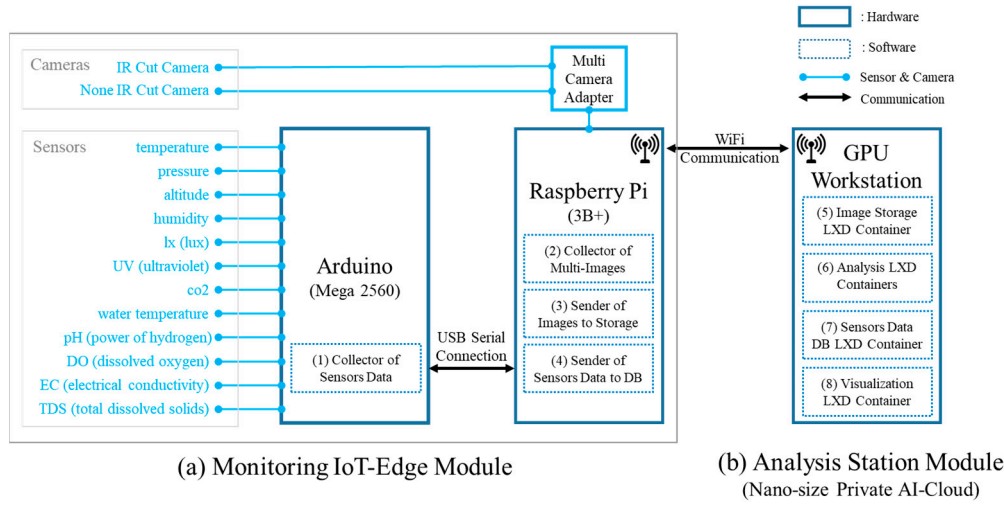

(a) Monitoring IoT-Edge Module    (b) Analysis Station Module
(Nano-size Private AI-Cloud)

**Figure 2.** The system module architecture.

3.2.1. Monitoring IoT-Edge Module

The IoT-Edge module has functions of collecting sensor data, transmitting sensing data to a database, taking multi-camera images and transmitting the images to a storage device. As shown in Figure 2a, the hardware of the IoT-Edge module consists of an Arduino module that acts as an IoT sensor hub and a Raspberry Pi module that acts as an edge device. Raspberry Pi's GPIO (General Purpose Input Output) does not support analog sensors because it does

not support ADCs (analog-to-digital converters). Additionally, the number of sensors that can be attached is limited because there are special-purpose GPIO pins. The Arduino module has 16 analog input pins and 54 digital input/output pins, so it can easily add sensors. The Arducam Multi Camera Adapter Module uses a single CSI camera port on the Raspberry Pi module to connect two cameras. It also uses three GPIOs on the Raspberry Pi module for multiplexing support. The Sensor Data Collector of Figure 2(1) collects 13 types of environmental data from eight sensors. Table 3 shows the sensors related to the data being collected. The Data Collector is programmed in C language using Sketch, an Arduino integrated development environment (IDE), to send the 13 types of sensing data to the IoT-Edge device every 0.5 s. The monitoring function of the IoT-Edge module is verified by collecting and transmitting data that are repeated every 0.5 s.

The Sensors Data Sender in Figure 2(4) stores the data collected through USB serial communication in the database of the GPU workstation. The Data Sender is programmed to receive sensing data from /dev/ttyUSB0 of the Raspberry Pi serial communication port using Python language and store it in the sensors table of the Maria database (i.e., Figure 2(7) Sensors Data DB LXD Container) of the workstation. The Multi Camera Image Collector in Figure 2(2) captures and stores strawberry images from IR cut (infrared cut-off filter) and non-IR cut cameras. The Python example program code of the Arducam Multi Camera Adapter (i.e., Multi_Camera_Adapter_V2.2_python [24]) was modified to capture strawberry images every 2 h with IR cut and non-IR cut cameras and store them in the IoT-Edge device. The Multi Camera Image Sender in Figure 2(3) saves the images stored on the IoT-Edge device to the workstation's image storage. The Image Sender was programmed in Bash shell script using the Linux command line utility SCP (Secure Copy) so that strawberry images stored on the Raspberry Pi module can be saved to the workstation's image storage every day.

### 3.2.2. Analysis Station Module

The analysis station module was designed with the concept of AI-Cloud, so if the system needs to be expanded, the server container can be flexibly and easily increased. In other words, the number of container servers based on virtualization can be easily increased whenever the strawberry cultivation area increases. As shown in Figure 2b, the analysis station module has functions, such as an image storage server in Figure 3c, analysis server in Figure 3d, database server in Figure 3b and visualization server in Figure 3e. The module was designed as a nano-sized private AI-Cloud to increase availability by separating it into containers for each function. This means it is separated into a virtualized container so that one of the analysis functions is shut down; as such, it cannot affect the operation of other functions. In addition, in order to increase the hardware resource pool efficiency and flexibility in relation to the functions of modules, it is composed of an infrastructure in which AI and cloud are hyper-converged. As shown in Figure 3, the servers for each function are containerized using Ubuntu's LXD [25]. LXD is a container hypervisor that Canonical made open source by improving the Linux container. Ubuntu version 18.04 was used as the operating system for the host server of the analysis station module, and as the operating system for the guest container server for each function.

The Database Server container in Figure 3b is a database server that stores sensor data from the Sensor Data Sender in Figure 2(4). MariaDB version 10.1.47 was installed in the container. To store sensor data, a database named "growing_environment" was created, and a table named "sensors" was created. Table 3 shows the schema of the sensor table by the "DESC sensor" query command. As shown in Table 4, the sensors table has 16 fields. The id field is a primary key, of which the number is automatically increased from 1. The time_sec field records a timestamp of the time point at which sensor data are stored in order to process them as a time series. The sensor field is used to identify IoT-Edge devices. In this work, since only one IoT-Edge device is used, 1 is recorded in the sensor field. The thirteen data types collected from the eight sensors in Table 1 (sensors) are stored fields in Table 3, from temperature to tds.

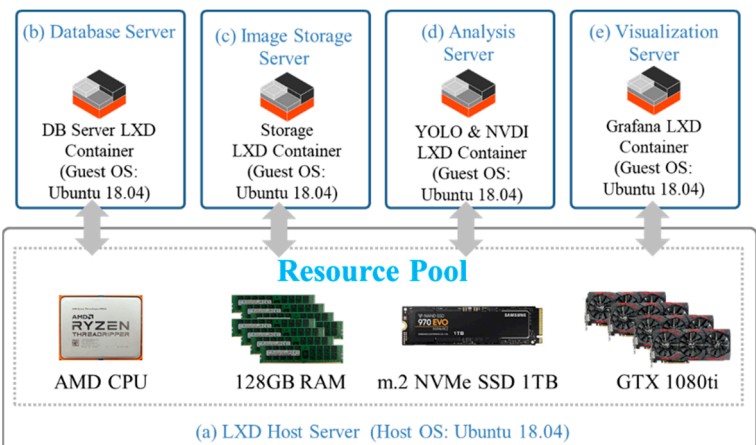

**Figure 3.** Nano-sized private AI-Cloud for the functions of analysis station module.

**Table 3.** The sensors table schema in the growing_environment database.

| Field | Type | Null | Key | Default | Extra |
|---|---|---|---|---|---|
| id | bigint (20) | NO | PRI | NULL | auto_increment |
| time_sec | timestamp | NO | | CURRENT_TIMESTAMP | on update CUR-RENT_TIMESTAMP |
| sensor | int (11) | YES | | NULL | |
| temperature | float | YES | | NULL | |
| pressure | float | YES | | NULL | |
| altitude | float | YES | | NULL | |
| humidity | float | YES | | NULL | |
| lux | float | YES | | NULL | |
| uv | float | YES | | NULL | |
| co2 | float | YES | | NULL | |
| co2status | char (2) | YES | | NULL | |
| wtemperature | float | YES | | NULL | |
| ph | float | YES | | NULL | |
| do | float | YES | | NULL | |
| ec | float | YES | | NULL | |
| tds | float | YES | | NULL | |

The Image Storage Server container in Figure 3c is an image storage server that stores strawberry pictures from the Multi Camera Image Sender in Figure 2(3). An image storage pool was created to separate the server container's storage pool. The name of the created image storage pool is img-storage, and the name of the existing storage pool is lxd-storage. The storage server container is attached by the img-storage pool, in which image data can be saved as a file in ext4 file system format [26,27].

The Analysis Server container in Figure 3d has the function of classifying strawberry images and calculating the normalized difference vegetation index (NDVI) [28]. Strawberry image classification is used to determine the harvest time.

The Visualization Server container in Figure 3e visualizes the sensor data of the Database Server container in Figure 3b and the results of strawberry object classification of the Analysis Server container in Figure 3d. Grafana version 7.1.5 was installed in the container for the visualization of sensor data and images. Ten fields were visualized by connecting Grafana with MariaDB's growing_environment database on the Database Server container. In addition, it visualizes the strawberry image of the Image Storage Server container in Figure 3c and the classified strawberry image of the Analysis Server container.

### 3.2.3. Data Handling of Analysis Server

NDVI can be used to analyze plant health by accurately indicating the state of chlorophyll by observing changes in near-infrared light compared to red light. The strawberry image classification function classifies the object of the strawberry image into six categories

according to the appearance maturity of the strawberry using a YOLO (You Only Look Once) algorithm. The YOLO algorithm enables end-to-end training and real-time speeds while maintaining high average precision. The algorithm is essentially a unified detection model without a complex processing pipeline that uses the whole image as the network input, which will be divided into an S × S grid. After selection from the network, the model directly outputs the position of the object border and the corresponding category in the output layer. However, the algorithm is not effective in detecting close objects and small populations. The versions of the YOLO algorithm consist of YOLO V1, YOLO V2 and YOLO V3. YOLO V1 transforms the target detection problem into a regression problem using a single convolutional neural network that extracts bounding boxes and class probabilities directly from the image. YOLO V2 is the improved version of YOLO V1. The modeling architecture and training model of YOLO V2 are proposed based on Darknet-19 and five anchor boxes. YOLO V3 integrates Darknet-19 from YOLO V2 to propose a new deeper and wider feature extraction network called Darknet-53. TinyYOLO is the light version of the YOLO. TinyYOLO is lighter and faster than YOLO while also outperforming other light models' accuracy [29,30].

Figure 4 shows a flowchart of the task of training a strawberry image classification function using the YOLO V3 algorithm and inferring strawberry image classification using the trained function. As shown in Figure 4a, 6156 strawberry images were classified into six categories by strawberry experts using YOLO Mark [31] for training the data. Figure 4b shows the inference of classification categories using the trained model and real strawberry image data. The categories consisted of immature, 30% mature, 50% mature, 60% mature, 80% mature and mature. Table 5 shows the classification categories of strawberry images and the number of training data sets. The maturation period of strawberries lasts up to 50 to 60 days in winter, and the maturation period gradually shortens as the temperature increases in spring. The category criteria in Table 4 are set for the spring season. The strawberry NDVI calculation function was calculated by using Equation (1) and the position coordinates of the strawberry object by the strawberry image classification function.

$$NDVI = \frac{NIR - RED}{NIR + RED} \tag{1}$$

where *NDVI* is the normalized differential vegetation index, *NIR* is a near-infrared value and *RED* is a red value. *NIR* and *RED* represent reflectivity measured in the near-infrared and red wavelength bands, respectively.

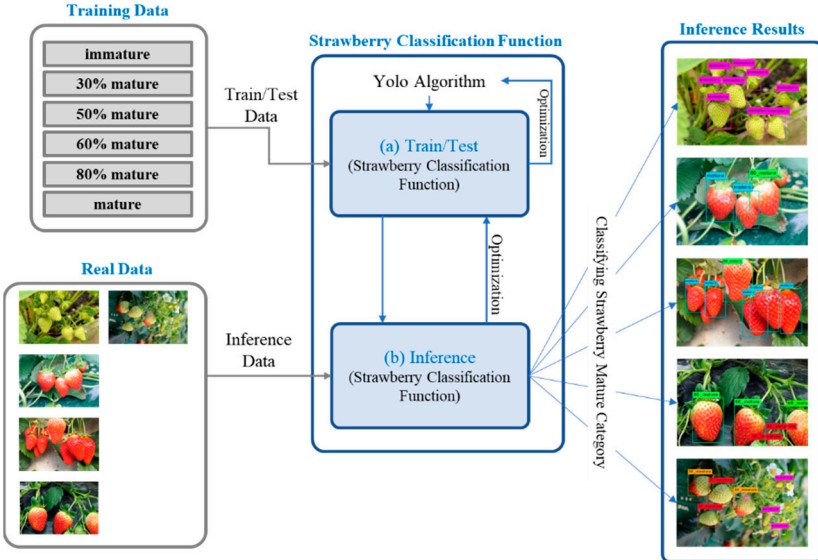

**Figure 4.** Strawberry harvest decision function workflow.

**Table 4.** Strawberry image classification categories and training data sets.

| Category | Label | 0 | 1 | 2 | 3 | 4 | 5 | Sum |
|---|---|---|---|---|---|---|---|---|
| | Meaning | Immature | 30% Mature | 50% Mature | 60% Mature | 80% Mature | Mature | |
| | | Harvest after 1 Month | Harvest after 3 Weeks | Harvest after 2 Weeks | Harvest after 1 Week | Harvest after 3 Days | Harvest after 1 Day | |
| Number of images | | 1005 | 1012 | 1017 | 1043 | 1038 | 1042 | 6156 |

## 4. Operational Use Case and Test

This section describes the operational use cases of the hydroponic strawberry monitoring and the tests of the harvest decision system. The monitoring system monitored environmental data for 5 months. Seolhyang strawberries [23] were grown for strawberry monitoring in a home hydroponic cultivation system. The monitoring system was located in our office with good natural light. April and May are actually the growing seasons of strawberries. From June, the temperature was so high that all the strawberries dried up and died. For this reason, only environmental information of the hydroponic cultivation system was monitored from June to August. The experiment data of the strawberry harvest decision system used 1575 strawberry photos taken using an MAPIR camera [32] at the Smart Berry Farm [33] in South Korea. In order to evaluate the classification accuracy of strawberry objects, 4329 strawberry objects included in the pictures were classified into the categories in Table 4 by strawberry-growing experts.

### 4.1. Database Server for Monitoring Data

Monitoring data collected by the IoT-Edge module were stored and managed in the MariaDB database server. The database server had a growing_environment database, which contained a sensor table. The sensor table was composed of the DB schema of Table 3. In the database server, 1,316,848 real data pieces, related to 13 categories of growing environmental data, were stored from 4 April 2020, to 31 August 2020. Table 5 shows the last 20 records stored in the sensors table using the query "select * from sensors order by id desc limit 20".

**Table 5.** Monitoring data of strawberry growing environment.

| Id | time_sec | Sensor | Temperature | Pressure | Altitude | Humidity | lux | uv | $CO_2$ | $CO_2$ Status | Wtemperature | ph | do | ec | tds |
|---|---|---|---|---|---|---|---|---|---|---|---|---|---|---|---|
| 93263 | 2020-08-26 13:55 | 1 | 30.97 | 992.31 | 175.8 | 28.43 | 357.5 | 0.16 | 515 | NR | 32.75 | 7.3 | 2.16 | 0.01 | 943.76 |
| 93264 | 2020-08-26 13:57 | 1 | 31.23 | 991.94 | 178.97 | 30.29 | 189.17 | 0.12 | 500 | NR | 32.44 | 7.31 | 2.13 | 0.01 | 943.76 |
| 93265 | 2020-08-26 13:59 | 1 | 31.24 | 991.98 | 178.63 | 30.33 | 197.5 | 0.12 | 497 | NR | 32.44 | 7.31 | 2.1 | 0.01 | 943.76 |
| 93266 | 2020-08-26 14:00 | 1 | 31.24 | 991.99 | 178.51 | 30.31 | 199.17 | 0.12 | 495 | NR | 32.44 | 7.31 | 2.07 | 0.01 | 943.76 |
| 93267 | 2020-08-26 14:00 | 1 | 31.22 | 992 | 178.42 | 30.22 | 200.83 | 0.16 | 495 | NR | 32.44 | 7.31 | 2.04 | 0.01 | 943.76 |
| 93268 | 2020-08-26 14:01 | 1 | 31.22 | 991.97 | 178.73 | 30.28 | 190 | 0.12 | 509 | NR | 32.44 | 7.31 | 2.02 | 0.01 | 943.76 |
| 93269 | 2020-08-26 14:03 | 1 | 31.17 | 992.01 | 178.39 | 30.48 | 203.33 | 0.12 | 508 | NR | 32.44 | 7.31 | 2.07 | 0.01 | 943.76 |
| 93270 | 2020-08-26 14:05 | 1 | 31.19 | 991.96 | 178.76 | 30.38 | 199.17 | 0.12 | 510 | NR | 32.44 | 7.31 | 2.05 | 0.01 | 943.76 |
| 93271 | 2020-08-26 14:05 | 1 | 31.2 | 991.99 | 178.54 | 30.38 | 197.5 | 0.12 | 510 | NR | 32.44 | 7.31 | 2.04 | 0.01 | 943.76 |
| 93272 | 2020-08-26 14:06 | 1 | 31.19 | 991.96 | 178.76 | 30.38 | 195 | 0.12 | 510 | NR | 32.44 | 7.31 | 2.05 | 0.01 | 943.76 |
| 93273 | 2020-08-26 14:06 | 1 | 31.21 | 991.99 | 178.55 | 30.43 | 191.67 | 0.12 | 508 | NR | 32.44 | 7.31 | 2.06 | 0.01 | 943.76 |
| 93274 | 2020-08-26 14:08 | 1 | 31.06 | 991.97 | 178.69 | 30.03 | 187.5 | 0.12 | 510 | NR | 32.44 | 7.31 | 2.03 | 0.01 | 943.76 |
| 93275 | 2020-08-26 14:10 | 1 | 30.92 | 991.92 | 179.13 | 30.31 | 183.33 | 0.12 | 495 | NR | 32.44 | 7.31 | 2.05 | 0.01 | 943.76 |
| 93276 | 2020-08-26 14:11 | 1 | 30.88 | 991.94 | 178.95 | 30.1 | 183.33 | 0.16 | 496 | NR | 32.44 | 7.31 | 2.14 | 0.01 | 943.76 |
| 93277 | 2020-08-26 14:11 | 1 | 30.85 | 992 | 178.41 | 30.46 | 184.17 | 0.12 | 498 | NR | 32.44 | 7.31 | 2.08 | 0.01 | 943.76 |
| 93278 | 2020-08-26 14:12 | 1 | 30.81 | 991.99 | 178.53 | 30.58 | 185 | 0.16 | 500 | NR | 32.44 | 7.31 | 2.03 | 0.01 | 943.76 |
| 93279 | 2020-08-26 14:14 | 1 | 30.88 | 991.91 | 179.24 | 30.88 | 193.33 | 0.13 | 499 | NR | 32.44 | 7.31 | 2.12 | 0.01 | 943.76 |
| 93280 | 2020-08-26 14:16 | 1 | 30.93 | 991.93 | 179.04 | 30.32 | 210.83 | 0.12 | 511 | NR | 32.44 | 7.31 | 2.05 | 0.01 | 943.76 |
| 93281 | 2020-08-26 14:16 | 1 | 30.96 | 991.91 | 179.24 | 30.75 | 216.67 | 0.16 | 511 | NR | 32.44 | 7.31 | 2.1 | 0.01 | 943.76 |
| 93282 | 2020-08-26 14:17 | 1 | 30.96 | 991.9 | 179.27 | 30.59 | 223.33 | 0.12 | 511 | NR | 32.44 | 7.31 | 2.13 | 0.01 | 943.76 |

### 4.2. Image Storage Server for Analysis Data

Strawberry pictures for strawberry harvest determination and NDVI calculation were taken by the IoT-Edge module and stored in the image storage server. In total,

3248 strawberry photos were taken and saved from 10 April 2020 to 25 August 2020. Half of the 3248 photos were taken with an IR cut camera and the other half with a non-IR cut camera. Figure 5 shows the pictures of strawberries stored on the image-storage server. These pictures were taken at intervals of 2 h. The photos named with camera-A were taken with the IR cut camera, and the photos named with camera-B were taken with the non-IR cut camera.

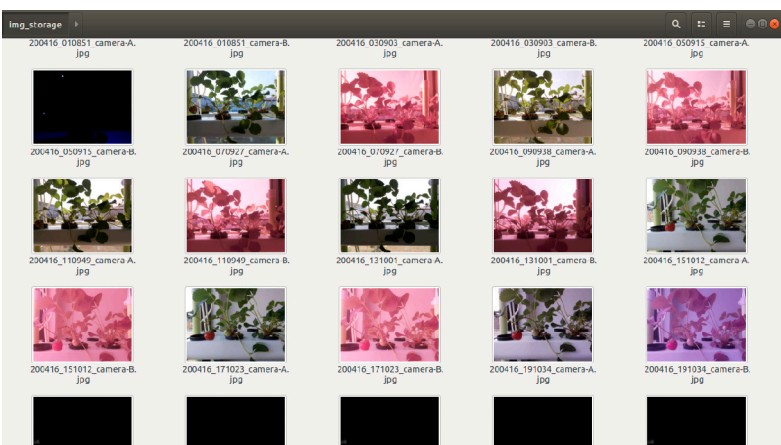

**Figure 5.** Strawberry photos stored on the Image Storage Server.

*4.3. Analysis Server for Strawberry Harvest Determination and NDVI Calculation*

In this subsection, the strawberry classification function and NDVI calculation tests are described. The classification accuracy was used to test the strawberry classification function. The accuracy rate was calculated by comparing the category classified by the strawberry classification model and the category classified by the expert. The classification model was created using the strawberry training data in Table 5 and the YOLO algorithm. The training of the classification model (i.e., YOLO V3) was repeated 50,020 times in 12 h and 5 min. Four NVidia GTX 1080ti graphic cards were used for training. Figure 6 shows an average loss rate of the training data of 0.0328 for 50,019 iterations in the YOLO V3 model.

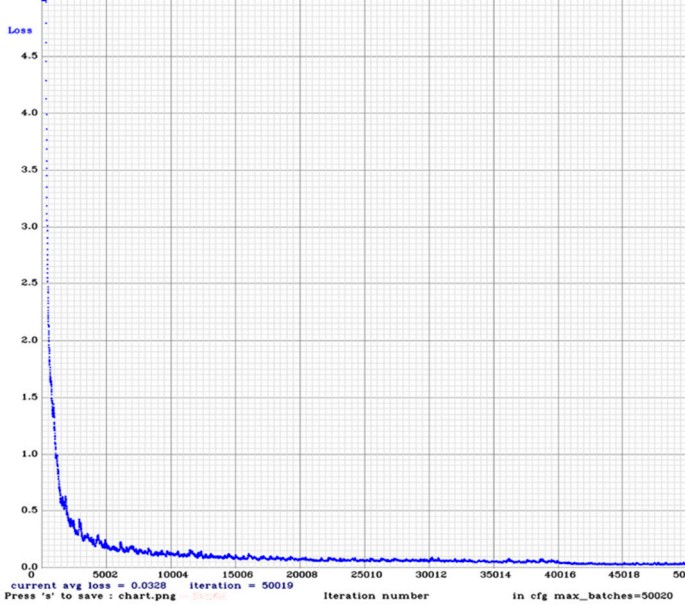

**Figure 6.** Average loss rate of training data for the strawberry classification model.

The training data in Table 4 were labeled to be optimized for the YOLO algorithm. For this reason, comparison with other methods is meaningless; therefore, in this paper, the YOLO V2, YOLO V3, TinyYOLO V2 and TinyYOLO V3 models are compared. YOLO V1 was excluded from the evaluation due to its low accuracy. YOLO models were created by transfer learning with the training data in Table 5 on the basic models of YOLO V2, YOLO V3, TinyYOLO V2, and TinyYOLO V3. The 4327 evaluation objects in 1575 strawberry photos consist of photos taken directly from strawberry farms and photos retrieved from a Google Images search. Figure 7 shows the comparison results of the average accuracy rates of the YOLO models for 4327 evaluation objects. In Figure 7, the average accuracy of YOLO V3 is approximately 3.667% higher than that of YOLO V2, 9.477% higher that of TinyYOLO V3 and 16.247% higher that of TinyYOLO V2. Due to the well-labeled training data and well-generated models, the accuracy rates of YOLO V3 and YOLO V2 are considered to be high. As the weight of the TinyYOLO model is smaller than that of YOLO, the training data are not well reflected in the generated model. As a result of analyzing 75 misclassified strawberry objects with YOLO V3, the objects overlap one another or the pictures are out of focus.

Figure 8 shows the process of calculating the NDVI value from a strawberry picture. As shown in Figure 8a, after selecting a strawberry object from a strawberry photo using a strawberry classification model, the coordinates of the object are extracted. As shown in Figure 8b, the NDVI values are calculated by using the coordinates of each object and Equation (1). Figure 8(1,2) were classified as label 1, which is 30% mature, by the strawberry classification model. Figure 8(3) was classified as label 4, which is 80% mature, by the strawberry classification model. In general, the NDVI value approaches −1 as the strawberry matures, and the NDVI value approaches 1 the more immature the strawberry is. The NDVI values of the strawberry objects in Figure 8(1,2) are very different. It is analyzed that the NDVI value is different because the intensity of the light source of the strawberry photos is different. In order to calculate an accurate NDVI value, an environment with a light source of constant intensity is required, such as a smart farm factory using LEDs.

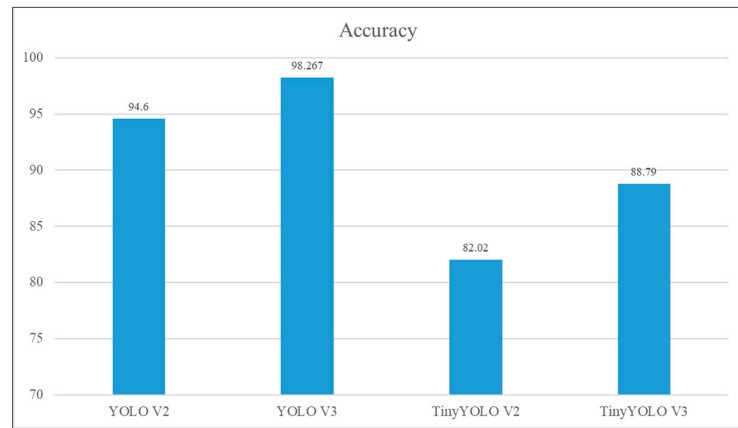

**Figure 7.** Result of comparison of the YOLO models for 4327 evaluation objects.

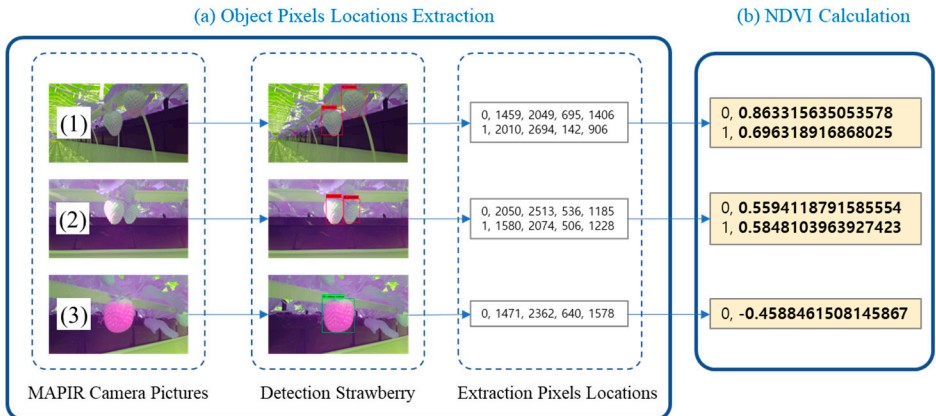

**Figure 8.** The process of calculating the NDVI value of the strawberry object.

### 4.4. Visualization Server for Strawberry Monitoring and Harvest Decision

On the visualization server, the monitoring and analysis results can be visualized with an integrated interface to use various basic data for growing strawberries. Figure 9 shows the visualization of the monitored strawberry environment data and the strawberry classification results with relation to harvest determination. Figure 9a shows the result of classifying strawberry photos by using the strawberry classification function of the analysis server and the IoT-Edge's IR cut camera. There are three strawberry objects in the photo. One object was classified as 30% mature, but the other two objects were not classified because they overlapped each other. Figure 9b shows the visualization of environmental data, such as humidity, temperature, water temperature, light, ultraviolet, $CO_2$, altitude, pressure, pH and dissolved oxygen, stored in the database server. Visualization of the NDVI values was excluded from the visualization server because the intensity of light constantly changed in the natural light environment.

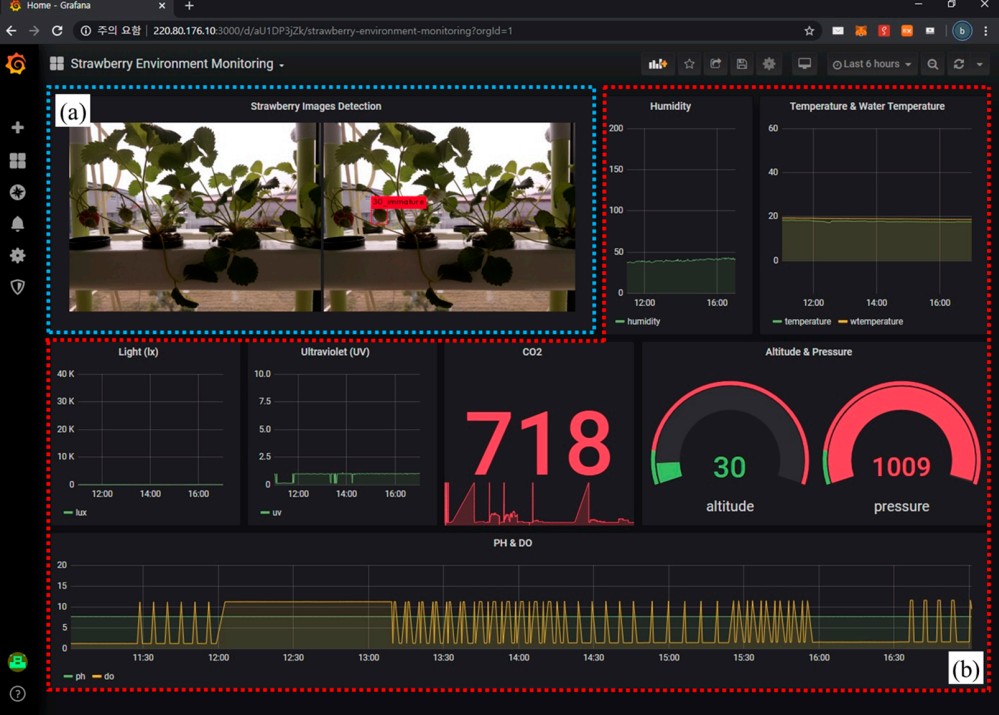

**Figure 9.** Visualization of the strawberry environment monitoring and the harvest decision.

## 5. Conclusions

In this study, we designed and implemented a system that monitors the strawberry hydroponic cultivation environment and determines the harvest time of strawberries. The proposed system uses an IoT-Edge module to collect strawberry hydroponic environment data and strawberry photos. The collected environmental data and strawberry photos are transferred to a nano-sized private AI-Cloud-based analysis station module and are visualized and determined when harvesting. The monitoring and analysis results visualized with an integrated interface provide a variety of basic data, such as varying yields, harvest times and pest diagnosis for strawberry cultivation. The proposed system was designed with the concept of an AI-Cloud, and the server container can be flexibly and easily increased if the system needs to be expanded. While growing Seolhyang strawberries in a home hydroponic cultivation system, the proposed monitoring system was tested by monitoring 1,316,848 actual environmental data pieces related to 13 data types over a period of 4 months. The proposed harvest decision system predicted the harvest time using 1575 strawberry pictures acquired from the Smart Berry Farm and a Google Images search and showed a high accuracy rate of 98.267%. As future research, we plan to study analysis methods that analyze the monitored strawberry growing environment data. In addition, we plan to study how the analysis results affect strawberry maturity.

**Author Contributions:** Conceptualization, S.P. and J.K.; methodology, S.P.; software, S.P.; validation, S.P. and J.K.; formal analysis, S.P.; investigation, S.P.; resources, S.P. data curation, S.P.; writing-original draft preparation, S.P.; writing-review and editing, J.K.; visualization, S.P. All authors have read and agreed to the published version of the manuscript.

**Funding:** This research was funded by Artificial Intelligence Graduate School Program (GIST).

**Data Availability Statement:** The data used to support the findings of this study are included within the article.

**Acknowledgments:** This work was supported by Institute of Information and Communications Technology Planning and Evaluation (IITP) grant funded by the Korean government (MSIT) (No.2019-0-01842, Artificial Intelligence Graduate School Program (GIST)).

**Conflicts of Interest:** The authors declare no conflict of interest.

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
