# Peer review of "Design and Implementation of a Hydroponic Strawberry Monitoring and Harvesting Timing Information Supporting System Based on Nano AI-Cloud and IoT-Edge"

_electronics, doi:10.3390/electronics10121400_

Round 1

Reviewer 1 Report

This paper presents a design and implementation of hydroponics strawberry monitoring and harvest decision system based on nano AI-Cloud and IoT-Edge. Based on the experiment results, the reviewer agrees that this design is functioning well. However, from the practical point of view, some issues have to be addressed.

  1. When does this system make harvest decision? Will it continuously report the decision results? Does the IoT Edge continuously report the collect data and images to the AI-Cloud?
  2. If the field of facility cultivation is large, could the camera working well? The authors should give your solution in such situation.
  3. When a harvest decision has been made, how does this system notify the farmer to harvest the matured strawberry in a efficient way?  

Author Response

Thank you for your kind and friendly review comment. We attach our response Word file.

Reviewer 2 Report

This paper presents a hydroponics strawberry monitoring system with an intelligent harvest decision algorithm. The work is interesting enough, and shows important experimental results, but lacks in technical depth and the actual contribution is not clearly portrayed by the authors, making the manuscript to read more as a engineering technical report or white paper, rather than a research article. The following comments are suggested for the author's consideration:

1. In page 1 it reads: "The global strawberry market size [1] is expected to reach $24.5 million by 2026, from $183.7 million in 2020." Authors should revise the numbers as they are describing a market growth but the market size by 2026 is lower than the one reported for 2020.

2. Please, define all acronyms (like YOLO) when used for the first time in the manuscript.

3. Section 2 describe previous related works. The references are recent and varied enough. However, the section describes only what previous works did. There is no a description of the limitations of those works that would help the readers to get a sense of how the proposed work in the manuscript advances the state-of-art. Also, it is suggested that special attention should be given to the works that are focused in monitoring and harvesting of strawberries. More information should be given what distinguishes the author's proposed system from such works, and a comparison could be given of what is being improved.

4. What are the technical reasons for using two microcontrollers? IoT sensing nodes are expected to be reduced in cost, size and weight, and more importantly, in power consumption. Having redundant components in the system (like two processing units) goes against those trends. Therefore, good technical reasons should be given as for why two microcontrollers are being used against selecting just one that is less limited in its capabilities.

5. Why does the Arduino senses the 13 data variables every 0.5 seconds? This seems like a lot of redundant data will be obtained, as most of the measured variables (temp., humidity, etc.) do not change that fast, much less in a controlled environment as a greenhouse or home system like the one is portrayed in the manuscript.

6. Table 3 and 1 has redundant information with respect to the sensors. It is suggested to remove Table 3.

7. It seems that the the classification algorithm relies solely on the captured images by the cameras. What is the purpose of the data taken for the other 13 variables? Why it is necessary to take that information?

8. The papers makes it clear that an extensive training was performed for the classification system, but it seems that there was not actual testing of the accuracy of the system once it has been trained. What is the error rate of the classification process once in operation? How many images (before the ones that were used for training) were used for testing?

Author Response

(The authors gave the same response as above.)

Reviewer 3 Report

Broad comments. The authors have made a concise overview of the topic and a brief reference to existing literature. They have indicated the main task of the paper among its motivation. Finally, they have pointed out the key message and the potential benefits of their work.

Specific comments. In general, the text is well structured and has clearly defined topics. The abstract is a good guide for what follows. Fundamental theory details that are needed are discussed but could be better analyzed. Analysis of the results is descriptive of the method's capability but could further be optimized. Concluding remarks are sufficient.

Some comments for improvement:

  1. The expected market size for 2026 in line 32 should be modified such that is larger than the relevant for 2020.
  2. Both references [1] and [2] could be moved to the end of the sentences.
  3. The second part of line 38 could be refined (better English to be used).
  4. Lines 54-56 could be refined to better describe the contribution of the work. An example could be the design and development of a system capable of real-time monitoring of the environment and supporting the decision-makers with enhanced information about harvesting timing.
  5. In line 58 please provide a reference for the concept
  6. While the last paragraph of section 1 is inclusive of author motivation and methodology, the innovation of the effort is hardly mentioned. Since this is a regular Article the innovation of the work compared to similar efforts should be highlighted in this paragraph and further supported by section 2.
  7. Information provided in lines 161- 165, seems to be irrelevant to the scope of the paper. Authors could consider removing them or include some extra info on the reasoning behind their selection (e.g. if these specific properties were judged to be important for the analysis).
  8. From figure 1(1) it is not evident how the cameras are mounted. Did the authors analyze the distance of installation and/or the view angle of the cameras? Would these parameters affect the precision of the methodology?
  9. In section 3.2 authors should explain the selection of a double Mcu-mpu architecture. It seems that either the Arduino with cam shield board (I2C) or the RPi through the GPIO header.
  10. Authors should provide more details about the accuracy of the sensors and justification of their choices. For example, BME280 seems to be out of date and presents much lower accuracy compared to its successor BME680.
  11. Is table 4 needed? Authors could consider replacing the table with a DB schema.
  12. While section 3 is very informative for the server architecture it also includes a description of the main data analysis pre-processing and methodology (lines252-276). Authors are suggested to move this information to a new section (eg data handling) and further expand it with specific details for the YOLO model used.
  13. Is figure 5 needed? Authors are encouraged to provide relevant information in a more tabular form.
  14. Authors could consider reducing the number of pictures included in figure 6.
  15. Lines 318 – 340 are describing the main outcome of the research. It is advisable to enrich this part of the paper with more justification regarding the quantification of the results (e.g. parametric experiments) and comparisons with different methods (if applicable). This way both the justification of specific selections and the innovation of the methodology could be highlighted.
  16. Authors could consider adding a paragraph with the usability of their methodology. While the method seems to be accurate enough it is based on an enormous amount of visual data (images) of a small number of subjects (strawberries). Could the trained model be applied on large scale?
  17. Authors could consider editing the language of the text. 

Author Response

(The authors gave the same response as above.)

Round 2

Reviewer 1 Report

The reviewer is satisfied with this revision. No further comments.

Author Response

Thank you for your kind and friendly review comment. We got the English editing services of MDPI for our paper. We have uploaded our edited paper.

Reviewer 2 Report

The authors have satisfactorily answered the reviewer's comments. No more further comments.

Author Response

(The authors gave the same response as above.)

Reviewer 3 Report

Thank you for the very nice feedback. 

Author Response

(The authors gave the same response as above.)
